# Sexual and reproductive health challenges among street adolescents in Sylhet city, Bangladesh: A cross-sectional study

Yeahyea Ahmed[1], Md Abdullah Saeed Khan[1], Laila Afroz[2], Mohammad Nurunnabi[3], Md Golam Abbas[1]*

**1** National Institute of Preventive and Social Medicine, Mohakhali, Dhaka, Bangladesh, **2** Institute of Leprosy Control and Hospital, Mohakhali, Dhaka, Bangladesh, **3** Sylhet Women's Medical College, Sylhet, Bangladesh

* abbasgolam@yahoo.com

## Abstract

### Background

Street adolescents often engage in early sexual activity, have multiple partners, and are at high risk of sexual abuse and exploitation. Despite the significance of this issue, there is a critical gap in understanding the sexual and reproductive health (SRH) needs, practices, and challenges of this marginalized population in Bangladesh, which this study aimed to explore.

### Methods

A cross-sectional study was conducted from August to December 2023, involving 311 street adolescents aged 16–19 years in Sylhet City Corporation, Bangladesh. Data were collected through face-to-face interviews using a semi-structured questionnaire. The questionnaire covered sociodemographic characteristics, pubertal changes, SRH status, and SRH-seeking behaviors.

### Results

Of all participants, 62.8% were males and 37.2% were females, with a mean age of approximately 17.3 years for both sexes. Sexual intercourse was reported by 32.56% of participants, with a significant gender disparity (76.79% females vs. 6.35% males, p<0.001). Contraceptive use was low at 14.14%, primarily condoms. Among female participants who ever had sex, 81.13% reported pregnancy, with 50.67% having had an abortion. Knowledge of sexually transmitted diseases (STDs) was alarmingly low, with only 2.33% aware of transmission methods. Less than half (45.97%) of participants utilized SRH services. Sexual abuse was reported by 61.79% of participants. Logistic regression revealed that younger age (Adjusted Odds Ratio [AOR]: 0.62, 95% CI: 0.48–0.79), being female (AOR: 9.10, 95% CI: 3.58–25.3) and longer

**Data availability statement:** All relevant data are within the manuscript and its Supporting information files.

**Funding:** The author(s) received no specific funding for this work.

**Competing interests:** The authors have declared that no competing interests exist.

duration of stay on the streets (AOR: 1.14, 95% CI: 1.02 to 1.27) were associated with higher odds of experiencing sexual abuse.

## Conclusion

Street adolescents in Sylhet City face severe SRH challenges, including high rates of sexual abuse, low contraceptive use, and limited STD knowledge, with significant gender disparities, which should be addressed through appropriate and urgent interventions.

---

## Introduction

Adolescence, defined by the World Health Organization as the period between 10–19 years [1], is a critical phase of human development marked by significant physical, emotional, and social changes. With 1.3 billion adolescents globally, comprising 16% of the world's population, this demographic represents a substantial and growing segment of society [2]. The transition from childhood to adulthood during adolescence is characterized by rapid physical maturation, cognitive development, and the emergence of sexual and reproductive capacities. During this period, adolescents develop the physical capacity for sexual activity and reproduction, alongside an increasing interest in sexuality and relationships. However, this biological readiness often precedes the development of full decision-making capacity and impulse control, creating a period of vulnerability [3,4]. This vulnerability is exacerbated by limited access to sexual and reproductive health (SRH) information and services, resulting in disproportionate SRH challenges among young people [2].

The situation is particularly dire for street adolescents, who face heightened risks and vulnerabilities. Globally, there are more than 100 million street youths, with a significant concentration in urban areas [5]. These young people are exposed to situations that make them exceptionally vulnerable to sexual and reproductive health problems. Street children often engage in early sexual activity, have multiple sexual partners, and are at high risk of sexual abuse and exploitation [6]. Studies have shown that HIV prevalence rates among street children are 10–25 times higher than among non-street adolescents [7].

In Bangladesh, the challenges faced by street adolescents are particularly acute. With over 600,000 street children, 75% of whom reside in Dhaka, these young people represent one of the most vulnerable populations in a country where 50% of the urban population lives below the poverty line [8]. The rapid urbanization and rural-urban migration in Bangladesh have contributed to an increasing number of "floating" people in urban areas [9], exacerbating the issue of street children.

Recent surveys have highlighted the dire circumstances of street children in Bangladesh. Approximately 30% lack basic amenities and reside in public spaces. Educational challenges are severe, with 72% unable to read or write. Alarmingly, 83% face abuse or harassment from pedestrians, and 39% from transport workers [10]. The sexual and reproductive health of street adolescents is not only a matter of

individual well-being but also has broader implications for public health and societal development [11]. High rates of unintended pregnancies, sexually transmitted infections, and other SRH issues among this population can strain healthcare systems [12] and perpetuate cycles of poverty and marginalization.

Sylhet is a metropolitan with distinct characteristics located in the most northeastern region of Bangladesh. Compared to other cities, it has a unique trend of urbanization within the context of distinct topological patterns, lush vegetation, torrential downpours, and a Sufi shrine-centric socio-economic growth [13]. Like other cities, the number of adolescents living on the street is increasing. Previous studies highlighted their extreme living conditions, food insecurity, risks of exploitation, and abuse [14,15]. However, very few studies explored their sexual and reproductive health needs and challenges. Given the unique challenges and vulnerabilities faced by street adolescents in Bangladesh [8,10,16] and the distinct geosocial context, we aimed to explore the SRH needs, practices, and challenges of this marginalized population in Sylhet city to inform the development of targeted interventions, policies, and services.

## Methods

### Study design, population, and period

This cross-sectional study was conducted among street adolescents in Sylhet City Corporation, Bangladesh from August to December, 2023. Participants were recruited from various locations, including Shahjalal Majar, Shahporan Majar, Zindabazar point, Medical area, Kin Bridge, Kadamtoli bus terminal, Sylhet railway station, and different roadsides in Sylhet City Corporation. The study included both male and female street adolescents aged 16–19 years. Exclusion criteria were severe illness, ages below 16 or above 19 years, unwillingness to participate, and mental illness.

### Sample size and sampling technique

The sample size was calculated using the formula $n = z^2pq/d^2$. Although the study explored various aspects of the sexual and reproductive health needs and challenges of the street children, to maximize the sample size based on available estimates from similar studies, the prevalence of sexual exposure (71.6%) among street children from the work of Habtamu and Adamu [17] was taken. Thereafter, with a 95% confidence interval and 5% precision, the final sample size was determined to be 312. With an assumed 10% non-response rate, the number of samples to be approached was determined to be 344. However, during the data collection period, a total of 330 participants could be approached, and 311 participants were included in the study. Informed consent could not be obtained from 29 participants (Fig 1). A convenient sampling technique was employed for data collection.

### Data collection instrument and procedure

A semi-structured questionnaire was developed in English, translated to Bengali, and back-translated to ensure accuracy. The questionnaire comprised four sections: (A) sociodemographic characteristics, (B) pubertal changes, (C) sexual and reproductive health status, and (D) sexual and reproductive health-seeking behaviors (**See S1 File for the questionnaire**). The instrument was pretested with 20 respondents, not included in the final sample. Data was collected through face-to-face interviews by the principal investigator himself. Each interview took approximately 15–20 minutes. Written informed consent was obtained from participants and legal guardians for those under 18 years old, with assent from the participants. The interview took place at their temporary living places on the streets. The interviews were conducted by the principal investigator himself, with only the participants present. This was to ensure that their responses to the sensitive questions were not affected by the presence of guardians or other acquaintances. However, interviews with female adolescents took place in the presence of a female individual (the principal investigator's wife) to ensure comfort, trust, cultural sensitivity, and honesty of responses. While asking about sexual abuse, the meaning of the term was clearly explained to the participants to provide a uniform understanding of the term. We defined sexual abuse as any form of

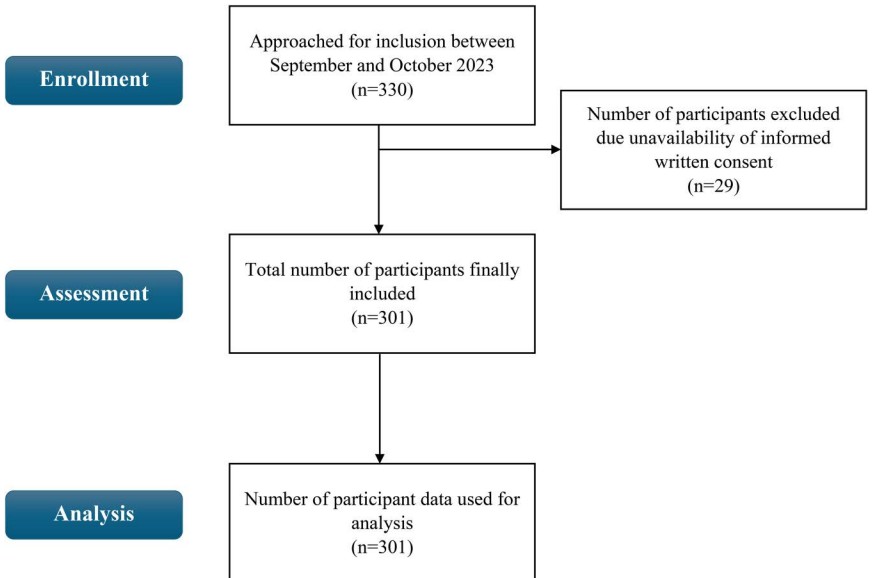

**Fig 1. Flow chart of participant selection.**

unwanted sexual contact, behavior, or attempt that makes an individual feel violated, uncomfortable, or coerced without their consent. This included acts such as inappropriate touching, coercion into sexual activities, verbal sexual harassment, or any other non-consensual sexual behavior.

## Data analysis

Data were processed, coded, and analyzed using RStudio Version 2024.09.0 and Microsoft Excel. Descriptive statistics were presented as frequency tables, bar charts, and pie charts. Pearson's chi-square test was used to examine associations between variables. Univariate and multivariable logistic regression analysis was performed to determine factors associated with sexual abuse among street adolescents. Factors for multivariable modeling were selected based on their relevance and significance in univariate analysis. A p-value of <0.05 was considered significant for all statistical tests.

## Ethical considerations

The study protocol was approved by the Institutional Review Board (IRB) of the National Institute of Preventive and Social Medicine (NIPSOM) (Memo no: NIPSOM/IRB/2023/06). Permission was obtained from the Councilor of different Wards in Sylhet City Corporation. Participants and their guardians (if available) were informed about the study's purpose before asking for consent and assent. Informed written assent was taken from participants aged <18 years, along with consent from their guardian (if available) or from the relevant Councilor if no legal guardian is available. For participants aged ≥18 years, informed written consent was taken. Confidentiality and anonymity were strictly maintained. All procedures were conducted following the latest guidelines of the Declaration of Helsinki.

## Results

The study included 301 street adolescents, with 62.8% (n = 189) males and 37.2% (n = 112) females (Fig 2). The mean age was similar for both sexes (17.35 ± 1.05 years for males, 17.32 ± 1.10 years for females). Significant differences were

**Sex distribution of the participants**

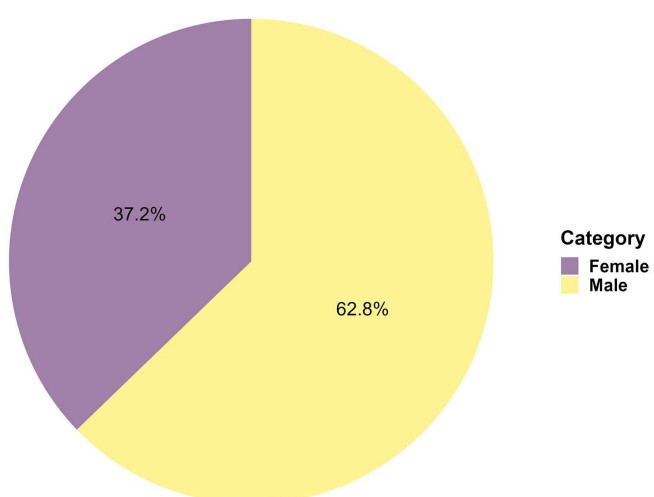

**Fig 2. Distribution of participants based on their sex.**

observed between males and females in marital status, education level, parental education, and occupations (p < 0.001 for all) (Table 1).

The median age of puberty onset was 12 years for both sexes. Among males, 96.83% had experienced wet dreams, with 28.57% using medication for it. For females, 98.21% had experienced menstruation, with a median age of menarche at 12 years. 20% of females reported menstrual problems, primarily pain (60.87%) (Table 2).

The study revealed that 32.56% of participants reported having had sexual intercourse, with 76.79% of females and 6.35% of males reporting sexual experiences (p < 0.001). Sexual health issues were reported by 6.64% of participants, with 8.99% of males and 2.68% of females reporting such issues (p = 0.033). Contraceptive use was reported by 14.14% of participants who ever had sexual intercourse, with 38.46% of males and 10.47% of females using contraceptives (p = 0.018). Among contraceptive users, 64.29% used condoms. Among female participants who ever had sex, 81.13% reported having conceived, and 50.67% reported having had an abortion. Knowledge about sexually transmitted diseases was limited, with 2.33% knowing how STDs spread and 3.65% aware of risk factors. Females showed slightly higher awareness in both categories (p = 0.012 and p = 0.022, respectively). Utilization of sexual and reproductive health services was reported by 45.97% of participants, with 50.00% of females and 14.29% of males using these services (p = 0.012). The main reasons for not using these services were lack of knowledge (43.28%), socio-cultural barriers (28.36%), and other reasons (26.87%), with significant differences between sexes (p < 0.001) (Table 3).

Sexual abuse was reported by 61.79% of participants, with 84.82% of females and 48.15% of males reporting such experiences (p < 0.001). The nature of abuse differed between genders (p < 0.001), with males primarily experiencing physical abuse (68.48%) and females experiencing more sexual (46.32%) and psychological (50.53%) abuse. The perpetrators of sexual abuse also differed between genders (p < 0.001). For males, peers were the primary abusers (79.12%), while for females, "others" were the main abusers (72.63%) (Table 4).

Logistic regression analysis revealed that younger age (OR: 0.62, 95% CI: 0.48–0.79, p < 0.001) and being female (OR: 9.10, 95% CI: 3.58–25.3, p < 0.001) were significantly associated with higher odds of experiencing sexual abuse. Longer duration of street stay was also associated with increased odds of abuse (OR: 1.14, 95% CI: 1.02–1.27, p = 0.022) (Table 5).

**Table 1. Sociodemographic characteristics of the participants by sex.**

| Characteristic | Sex | | | p-value |
|---|---|---|---|---|
| | Overall N = 301[1] | Female N = 112[1] | Male N = 189[1] | |
| **Age (years)** | 17.34 (1.07) | 17.32 (1.10) | 17.35 (1.05) | 0.798[2] |
| **Age groups (years)** | | | | 0.341[3] |
| 16 to 17 | 164 (54.49) | 65 (58.04) | 99 (52.38) | |
| 18 to 19 | 137 (45.51) | 47 (41.96) | 90 (47.62) | |
| **Marital Status** | | | | <0.001[4] |
| Unmarried | 210 (69.77) | 38 (33.93) | 172 (91.01) | |
| Married | 23 (7.64) | 11 (9.82) | 12 (6.35) | |
| Divorced | 7 (2.33) | 6 (5.36) | 1 (0.53) | |
| Separated | 43 (14.29) | 39 (34.82) | 4 (2.12) | |
| Widow | 18 (5.98) | 18 (16.07) | 0 (0.00) | |
| **Level of education** | | | | <0.001[3] |
| No formal education | 181 (60.13) | 53 (47.32) | 128 (67.72) | |
| Primary | 120 (39.87) | 59 (52.68) | 61 (32.28) | |
| **Father's level of education** | | | | <0.001[4] |
| No formal education | 215 (71.43) | 57 (50.89) | 158 (83.60) | |
| Primary | 83 (27.57) | 52 (46.43) | 31 (16.40) | |
| Higher education | 3 (1.00) | 3 (2.68) | 0 (0.00) | |
| **Mother's level of education** | | | | <0.001[4] |
| No formal education | 250 (83.06) | 81 (72.32) | 169 (89.42) | |
| Primary | 49 (16.28) | 29 (25.89) | 20 (10.58) | |
| Higher education | 2 (0.66) | 2 (1.79) | 0 (0.00) | |
| **Father's occupation** | | | | <0.001[4] |
| Agricultural worker | 7 (2.33) | 0 (0.00) | 7 (3.70) | |
| Business | 13 (4.32) | 0 (0.00) | 13 (6.88) | |
| Day labor | 89 (29.57) | 15 (13.39) | 74 (39.15) | |
| Others | 164 (54.49) | 87 (77.68) | 77 (40.74) | |
| Unemployed | 28 (9.30) | 10 (8.93) | 18 (9.52) | |
| **Mother's occupation** | | | | <0.001[3] |
| Home maker | 71 (23.59) | 9 (8.04) | 62 (32.80) | |
| Others | 116 (38.54) | 45 (40.18) | 71 (37.57) | |
| Unemployed | 114 (37.87) | 58 (51.79) | 56 (29.63) | |
| **Living status** | | | | <0.001[3] |
| Alone | 246 (81.73) | 64 (57.14) | 182 (96.30) | |
| Living with mother | 55 (18.27) | 48 (42.86) | 7 (3.70) | |
| **Type of family** | | | | >0.999[4] |
| Joint family | 3 (1.00) | 1 (0.89) | 2 (1.06) | |
| Nuclear family | 298 (99.00) | 111 (99.11) | 187 (98.94) | |
| **Number of siblings** | 5.00 (3.00, 6.00) | 5.00 (3.50, 6.00) | 4.00 (3.00, 5.00) | 0.004[5] |
| **Daily income (BDT)** | 127.57 (40.48) | 102.58 (30.70) | 142.38 (38.29) | <0.001[2] |
| **Source of earning** | | | | <0.001[4] |
| Begging | 228 (75.75) | 111 (99.11) | 117 (61.90) | |
| Collecting old materials | 43 (14.29) | 1 (0.89) | 42 (22.22) | |
| Porter | 6 (1.99) | 0 (0.00) | 6 (3.17) | |
| Pushing rickshaw | 18 (5.98) | 0 (0.00) | 18 (9.52) | |

*(Continued)*

**Table 1.** (Continued)

| Characteristic | | Sex | | p-value |
|---|---|---|---|---|
| | Overall N = 301[1] | Female N = 112[1] | Male N = 189[1] | |
| Selling old stuff | 6 (1.99) | 0 (0.00) | 6 (3.17) | |
| **Daily expenditure (BDT)** | 131.56 (37.26) | 112.41 (25.83) | 142.91 (38.39) | **<0.001[2]** |
| **Duration of stay on the streets (years)** | 5.00 (4.00, 7.00) | 5.00 (3.00, 6.50) | 6.00 (4.00, 8.00) | **0.035[5]** |
| **Presence of a sanitary latrine at the place of stay** | 293 (97.34) | 109 (97.32) | 184 (97.35) | >0.999[4] |

[1]Mean (SD); n (%); Median (Q1, Q3).

[2]Welch Two Sample t-test.

[3]Pearson's Chi-squared test.

[4]Fisher's exact test.

[5]Wilcoxon rank sum test.

**Table 2. Puberty and menstruation related characteristics of the participants.**

| Characteristic | N = 301[1] |
|---|---|
| **Male participants** | |
| Age at puberty (years) | 12.00 (12.00, 13.00) |
| Ever experienced wet dream | 183 (96.83) |
| Ever used medication for wet dream | 54 (28.57) |
| Source of medication for wet dream | |
| Herbal | 51 (94.44) |
| Pharmacy | 2 (3.70) |
| Homeopath | 1 (1.85) |
| **Female participants** | |
| Age at menstruation (years) | 110 (98.21) |
| Regularity of menstruation | |
| Regular | 88 (80.00) |
| Irregular | 22 (20.00) |
| Age at menarche (years) | 12.00 (12.00, 13.00) |
| Ever experienced menstrual problem | 22 (20.00) |
| Type of menstrual problem experienced | |
| Too much pain | 14 (60.87) |
| Too much bleeding and pain | 6 (26.09) |
| Too much bleeding | 3 (13.04) |
| Type of absorbent used for menstruation | |
| Cloths | 101 (91.82) |
| Sanitary pad | 9 (8.18) |
| Reasons for not using sanitary napkin during menstruation | |
| Costly | 98 (96.08) |
| Don't know about it | 4 (3.92) |

[1]Median (IQR); n (%).

**Table 3. Sex and sexual and reproductive health related information of the participants by sex.**

| Characteristic | Overall, N = 301[1] | Sex Male, N = 189[1] | Female, N = 112[1] | p-value |
|---|---|---|---|---|
| **Ever had sexual intercourse** | 98 (32.56) | 12 (6.35) | 86 (76.79) | **<0.001[2]** |
| **Ever had any sexual problems or illnesses** | 20 (6.64) | 17 (8.99) | 3 (2.68) | **0.033[2]** |
| **Uses any contraceptive*** | 14 (14.14) | 5 (38.46) | 9 (10.47) | **0.018[3]** |
| **Contraceptive method used*** | | | | 0.086[3] |
| Condom | 9 (64.29) | 5 (100.00) | 4 (44.44) | |
| Pill | 5 (35.71) | 0 (0.00) | 5 (55.56) | |
| **Ever conceived*** | 88 (81.48) | NA | 86 (81.13) | |
| **Ever had an abortion*** | 39 (51.32) | NA | 38 (50.67) | |
| **Ever faced problems in sex life** | 1 (0.33) | 0 (0.00) | 1 (0.89) | 0.372[3] |
| **Ever sought treatment for sex problem** | 13 (4.32) | 7 (3.70) | 6 (5.36) | 0.562[3] |
| **Place from where treatment for sexual and reproductive problem was sought** | | | | 0.133[3] |
| Govt. hospital | 281 (93.36) | 172 (91.01) | 109 (97.32) | |
| Medicine shop/pharmacy | 12 (3.99) | 10 (5.29) | 2 (1.79) | |
| Others | 5 (1.66) | 5 (2.65) | 0 (0.00) | |
| Private hospital | 2 (0.66) | 1 (0.53) | 1 (0.89) | |
| Community hospital | 1 (0.33) | 1 (0.53) | 0 (0.00) | |
| **Knows how sexually transmitted disease spreads** | 7 (2.33) | 1 (0.53) | 6 (5.36) | **0.012[3]** |
| **Knows about risk factors of sexually transmitted diseases** | 11 (3.65) | 3 (1.59) | 8 (7.14) | **0.022[3]** |
| **Ever used sexual and reproductive health services** | 57 (45.97) | 2 (14.29) | 55 (50.00) | **0.012[2]** |
| **Reasons for not using sexual and reproductive health services** | | | | **<0.001[3]** |
| Lack of knowledge | 29 (43.28) | 11 (91.67) | 18 (32.73) | |
| Socio-cultural barrier | 19 (28.36) | 0 (0.00) | 19 (34.55) | |
| Others | 18 (26.87) | 0 (0.00) | 18 (32.73) | |
| Lack of adolescent-friendly services | 1 (1.49) | 1 (8.33) | 0 (0.00) | |

[1]n (%).

[2]Pearson's Chi-squared test.

[3]Fisher's exact test.

*Among those who ever had sexual intercourse.

**Table 4. Sexual abuse-related information of the respondents by sex.**

| Characteristic | Overall, N = 301[1] | Sex Male, N = 189[1] | Female, N = 112[1] | p-value |
|---|---|---|---|---|
| **Ever experienced sexual abuse** | 186 (61.79) | 91 (48.15) | 95 (84.82) | **<0.001[2]** |
| **Nature of sexual abuse experienced** | | | | **<0.001[2]** |
| Physical | 66 (35.29) | 63 (68.48) | 3 (3.16) | |
| Sexual | 63 (33.69) | 19 (20.65) | 44 (46.32) | |
| Psychological | 58 (31.02) | 10 (10.87) | 48 (50.53) | |
| **Person by whom the sexual abuse occurred** | | | | **<0.001[3]** |
| Peers | 96 (51.61) | 72 (79.12) | 24 (25.26) | |
| Others | 87 (46.77) | 18 (19.78) | 69 (72.63) | |
| Relatives | 3 (1.61) | 1 (1.10) | 2 (2.11) | |

[1]n (%).

[2]Pearson's Chi-squared test.

[3]Fisher's exact test.

**Table 5. Univariate and multivariable logistic regression analysis exploring factors associated with sexual abuse of street adolescents.**

| Characteristic | Univariate Models | | | Multivariate Model | | |
|---|---|---|---|---|---|---|
| | OR[1] | 95% CI[1] | p-value | AOR[1] | 95% CI[1] | p-value |
| **Age (years)** | 0.66 | 0.53 to 0.83 | **<0.001** | 0.62 | 0.48 to 0.79 | **<0.001** |
| **Sex** | | | | | | |
| Male | — | — | | — | — | |
| Female | 6.02 | 3.41 to 11.1 | **<0.001** | 9.10 | 3.58 to 25.3 | **<0.001** |
| **Marital status** | | | | | | |
| Unmarried | — | — | | — | — | |
| Ever married | 2.70 | 1.57 to 4.81 | **<0.001** | 1.05 | 0.47 to 2.31 | 0.903 |
| **Level of education** | | | | | | |
| No formal education | — | — | | — | — | |
| Primary | 1.33 | 0.83 to 2.16 | 0.241 | 0.90 | 0.51 to 1.58 | 0.710 |
| **Living status** | | | | | | |
| Lives alone | — | — | | — | — | |
| Lives with mother | 2.29 | 1.20 to 4.64 | **0.016** | 0.77 | 0.31 to 1.90 | 0.576 |
| **Type of family** | | | | | | |
| Joint family | — | — | | — | — | |
| Nuclear family | 3.27 | 0.31 to 70.9 | 0.335 | 3.24 | 0.20 to 104 | 0.440 |
| **Daily income (BDT)** | 0.99 | 0.99 to 1.00 | **0.005** | 1.00 | 1.00 to 1.01 | 0.491 |
| **Duration of stay on streets (years)** | 1.07 | 0.97 to 1.17 | 0.164 | 1.14 | 1.02 to 1.27 | **0.022** |

[1]OR = Odds Ratio, AOR = Adjusted Odds Ratio, CI = Confidence Interval.

## Discussion

This cross-sectional study provides valuable insights into the sexual and reproductive health-related behaviors among street adolescents in Sylhet City, Bangladesh. The findings highlight significant gender disparities and vulnerabilities faced by this marginalized population.

The demographic profile of our study participants aligns with previous research [10], showing a predominance of males (62.8%) among street adolescents. This gender imbalance is consistent across studies [18–20]. Gender roles and the nature of the studies may somewhat explain the difference. For instance, male adolescents are more likely to work on the street, while female adolescents are likely to work at homes [21]. Moreover, most studies focus on male street children leading to a possible underestimation of female counterparts. However, it may also reflect differing vulnerabilities or social factors leading to street life, which warrants further investigation.

Our study revealed a high prevalence of illiteracy (60.1%) among street adolescents, which is slightly higher than the 49.6% reported by Chowdhury S. *et al.* [16] and lower than that found in the national survey on street children by the Bangladesh Bureau of Statistics [10]. The survey also revealed that around 64% of street children get enrolled in school, but only 22% can continue their education despite many obstacles. This underscores the critical need for educational interventions tailored to this population, as education could be a key factor in breaking the cycle of homelessness and improving overall life outcomes.

The economic status of street adolescents in our study was precarious, with a majority earning less than 100 TK per day, primarily through begging (73.8%). This finding differs slightly from previous studies, where Chowdhury S. *et al.* [16] reported 65.6% of respondents earning less than 100 TK per day. The predominance of begging as a source of income indicates a lack of alternative opportunities, education, and initiatives to rehabilitate this part of the population. Alongside

education, the government, through partnership with different non-government organizations, should arrange vocational training and create job opportunities for this group of adolescents to ensure their social security and rights.

Regarding sexual and reproductive health, our study found that 32.6% of adolescents reported previous sexual initiation, which is considerably lower than the 70.6% reported by Ewunetie *et al.* [12] and 53.9% reported by Simret *et al.* [22] in Ethiopia. This discrepancy could be due to educational and cultural differences, underreporting, or variations in study populations. For instance, studies conducted among school-going adolescents found that 17.2% in Tigray [23] and 36% in Harari region [24] schools in Ethiopia had experienced sexual intercourse, a figure lower than that of street adolescents. Considering the religious affiliation of the majority in the country, the sexual initiation rate appears to be much higher than what is expected to be found because of the religious restrictions [25,26]. This underscores that the reason for sexual initiation at this age is most likely to be coercive. The high prevalence of sexual abuse (61.79%), particularly among females (84.82%), found in this study also supports this reasoning. Organizations involved in women's rights movements and reproductive and sexual health services should take special initiatives to ensure the safety of adolescent street girls and justice to those who are the victims of exploitation. These initiatives should include plans for long-term physical, mental, and socio-economic rehabilitation of the sufferers.

The low rate of contraceptive use (14.1%) among sexually active adolescents is alarming and significantly lower than the 46.6% consistent condom use reported by Ewunetie *et al.* [12]. This gap in contraceptive use highlights the urgent need for comprehensive sexual education and improved access to contraceptive methods. It also explains the high proportion of conception (81.5% and abortion (51.3%) among female respondents in the study, which is higher than previous findings by Habtamu *et al.* [17], who reported 70.4% pregnancy rates and 85.5% abortion rates. These figures underscore the critical need for targeted interventions to prevent unwanted pregnancies and provide comprehensive post-conception care.

Knowledge about sexually transmitted diseases (STDs) was alarmingly low in our study population, with almost all respondents lacking knowledge about STD transmission and risk factors. This is in stark contrast to findings by Hossain *et al.* [27] among Bangladeshi women, where approximately 71% of respondents had some knowledge regarding STD transmission. Even women living in the urban slums had a fair proportion of knowledge (68.3%) regarding STD [28]. This highlights the urgent need for improved sexual health education among street adolescents in Sylhet city.

The utilization of sexual and reproductive health services was low (45.97%), with significant gender differences. This finding is lower than that reported by Bwamale *et al.* [29] (61.99%) in Ugandan street children. Barriers such as lack of knowledge and socio-cultural factors [30,31] continue to hinder access to these essential services.

We found that adolescent girls with a younger age and a shorter duration of stay on the street were significantly more likely to be sexually abused, indicating that young girls who are new to the streets are vulnerable to sexual exploitation. Kimberly and colleagues pointed out that adolescents who run away from home at a younger age, females, those having increased sexual activity, and who have trouble finding food and shelter are more likely to be the subject of sexual victimization [32,33]. Stakeholders must act on areas like sexual education and social safety nets tailored to street adolescents to prevent the sexual exploitation of those living on the streets.

The National Child Policy was instituted in 2011 in Bangladesh. Although the policy talks about the development, nutrition, safety, and health of children and adolescents, only one clause (no 6.2.3) specifically mentions street children. Given the persistent burden and challenges of street children and adolescents, it's time that the policies are updated to incorporate protective and rehabilitation measures for street children and adolescents, which, in addition to other rights, should ensure their sexual and reproductive safety and health needs.

## Limitations and strengths

This study has several limitations. The cross-sectional design prevents the establishment of causal relationships. The study used convenience sampling as there was no updated information on the number of street children currently living

in the city. Although this may limit the generalizability of the findings, to ensure variations in sampling, we collected data from different parts of the city where street children mostly stayed at night. The sensitive nature of the topics may have led respondents to underreport certain behaviors or experiences. The principal investigator himself was involved in the data collection to ensure the proper readout of the questionnaire with explanations as needed. However, the investigator objectively recorded the responses to prevent potential interviewer bias. Despite these limitations, the study has notable strengths. It provides valuable insights into a hard-to-reach population, addressing a significant gap in the literature on sexual and reproductive health behaviors among street adolescents in Bangladesh. The use of face-to-face interviews allowed for more in-depth data collection, and the inclusion of both male and female participants provides a comprehensive view of gender-specific issues.

## Conclusion

This study provides critical insights into the sexual and reproductive health behaviors of street adolescents in Sylhet City, Bangladesh. The findings highlight significant vulnerabilities, including low contraceptive use, high rates of sexual abuse, and limited knowledge of STDs. Gender disparities were evident across multiple dimensions. The results underscore the urgent need for targeted interventions to improve sexual education, increase access to reproductive health services, and address the specific challenges faced by female street adolescents. Future policies and programs should focus on providing comprehensive support to this marginalized population, with particular emphasis on preventing sexual exploitation and improving overall health outcomes. Further research is needed to develop effective, context-specific strategies for supporting street adolescents' sexual and reproductive health.

## Supporting information

**S1 File. Questionnaire.**
(DOCX)

**S2 File. Anonymized dataset.**
(CSV)

## Acknowledgments

The authors acknowledge the respondents for their voluntary participation in the study.

The Large Language Model Claude Sonnet 3.0 was used to improve the grammar and flow of writing of the manuscript. The authors extensively reviewed the manuscript after modification using AI to check for the correctness of description and interpretation.

## Author contributions

**Conceptualization:** Yeahyea Ahmed, Md Golam Abbas.

**Data curation:** Md Abdullah Saeed Khan.

**Formal analysis:** Yeahyea Ahmed, Md Abdullah Saeed Khan.

**Investigation:** Yeahyea Ahmed, Laila Afroz, Mohammad Nurunnabi.

**Methodology:** Yeahyea Ahmed, Md Golam Abbas.

**Project administration:** Yeahyea Ahmed, Md Golam Abbas.

**Resources:** Yeahyea Ahmed, Md Abdullah Saeed Khan, Laila Afroz, Mohammad Nurunnabi.

**Software:** Md Abdullah Saeed Khan.

 

**Supervision:** Md Golam Abbas.

**Visualization:** Md Abdullah Saeed Khan.

**Writing – original draft:** Yeahyea Ahmed, Md Abdullah Saeed Khan.

**Writing – review & editing:** Yeahyea Ahmed, Md Abdullah Saeed Khan, Laila Afroz, Mohammad Nurunnabi, Md Golam Abbas.

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
