## [Decision Letter · Decision Letter 0]

10 Sep 2025

Dear Dr.  Abbas,

We look forward to receiving your revised manuscript.

Kind regards,

Christina M. Roberts, M.D., M.P.H.

Academic Editor

PLOS ONE

**Comments from peer reviewers and academic editor**

Reviewer 1:

Thank you so much for providing me the opportunity to review the paper titled, “Sexual and Reproductive Health Challenges Among Street Adolescents in Sylhet City, Bangladesh: A Cross-Sectional Study”. The paper has analyzed an interesting issue, focusing on sexual reproductive health of street adolescents. The paper is well-written, including background, methodology, and results. However, I am suggesting some minor modification to the paper to make it more readable.

First, why the authors have chosen Sylhet city as study site, need explanation. As the case of Sylhet city might not be identical with other cities in Bangladesh regarding adolescent’s experience. It would be better to describe the context of Sylhet city in relation to other cities in Bangladesh.

Second, in the discussion section, the authors described and compared the results with other studies. However, they did not report convincingly the implications for policy and practice. For example, female adolescents are exposed to more sexual abuse than their male counterpart. So what? What policies and practices can be adopted in the family, community or state level to reduce risk and protect them? What is the implication for broader audiences other than the city studied or beyond Bangladesh? What are the socio-cultural barriers for not using SRH service?

Third, the authors used inferential statistics with a convenience sampling. They should report the reasons for not using probability sampling.

Fourth, there is an inconsistency in reporting results. (line no 146-47) Longer duration of street stay was also associated with 174 increased odds of abuse (OR: 1.14, 95% CI: 1.02-1.27, p=0.022) (Table 5). ) (Line 229-231) We found that adolescent girls with a younger age and a shorter duration of stay on the street were 230 significantly more likely to be sexually abused/ Which one is correct?

Reviewer 2:

I think the authors rightly addressed that the street adolescents who engage in early sexual activity, have multiple partners, and are at high risk of sexual abuse and exploitation and tried to explore the significance of this issue in Sylhet region.

Reviewer 3:

This manuscript addresses a significant and underexplored topic in public health. The objective to understand the sexual and reproductive health (SRH) needs, practices, and challenges of the study population is highly relevant and commendable. However, major methodological and analytical concerns currently preclude the acceptance of the manuscript in its present form. Addressing these issues is essential.

Major Comments:

• Sample Size Justification: The sample size was calculated using the prevalence of 'sexual exposure' (71.6%). However, the study's primary objective is to understand a broad range of SRH needs, practices, and challenges, which are not considered for the sample size calculation. Please clarify the rationale for the sample size calculation.

• Risk of Investigator Bias: The description of data collection is unclear. The term "researcher" is not defined, and the involvement of the investigator's wife is noted. As the study team was fully aware of the research objectives, their direct role in conducting sensitive face-to-face interviews introduces a significant risk of observer and interviewer bias.

• Analytical Approach and Statistical Rigor: The analytical strategy requires review. The stratification of results in Tables 1, 3, and 4 by participant sex is not clearly justified by the study objectives. If sex comparison is not a central aim, this stratification is unnecessary.

• The measuring "sexual abuse" are not clearly described. The construction of the multivariate model (Table 5) needs further revised. The model uses current factors (e.g., age, marital status, education at the time of survey) to predict a lifetime history of sexual abuse. This creates a temporal fallacy where a current state is incorrectly presented as influencing a past event. The interpretation of current age as a protective factor (AOR: 0.62) against historical abuse is not logically correct. Similar to other factors as well.

Editor Comments:

You used a semi-structured questionnaire for your research. This implies use of both qualitative and quantitative methods in your study. The qualitative methods and results for your study need additional elaboration. The Equator Network guideline for reporting the results of qualitative health research provides a useful guideline for describing the results of studies like yours.

O'Brien BC, Harris IB, Beckman TJ, Reed DA, Cook DA. Standards for reporting qualitative research: a synthesis of recommendations. Acad Med. 2014;89(9):1245-1251.

https://www.equator-network.org/reporting-guidelines/srqr/

Providing this additional information may be useful in addressing the concerns expressed by Reviewer 3 and, to a lesser extent, Reviewer 1.

While elaborating on the qualitative aspects of your study, I would like you to elaborate on the following items:

• methods for collecting and interpreting the qualitative data in your study,

• elaboration of the major themes developed, for example "sexual abuse",

• Justification for the inferential statistics used to analyze your qualitative data.

I look forward to your reply.

Christina Roberts, MD MPH

Journal Requirements:

Additional Editor Comments:

Thank you for the opportunity to review your manuscript, “Sexual and Reproductive Health Challenges Among Street Adolescents in Sylhet City, Bangladesh: A Cross-Sectional Study”. I commend you (and your wife) for tackling an under investigated topic with a hard to reach population. The novelty of your research topic is a great strength of your study. However, your peer reviewers and I have a few concerns we would like you to address to strengthen your manuscript before it is acceptable for publication.

1. I have included the comments from our reviewers to inform your efforts if you choose to move forward the revising your paper for publication in PLOS One.

Kind regards,

Christina M. Roberts, M.D., M.P.H.

Academic Editor

PLOS ONE

Reviewer's Responses to Questions

**Comments to the Author**

1. Is the manuscript technically sound, and do the data support the conclusions?

Reviewer #1: Yes

Reviewer #2: Yes

Reviewer #3: No

2. Has the statistical analysis been performed appropriately and rigorously?

Reviewer #1: Yes

Reviewer #2: Yes

Reviewer #3: No

3. Have the authors made all data underlying the findings in their manuscript fully available?

Reviewer #1: Yes

Reviewer #2: Yes

Reviewer #3: Yes

4. Is the manuscript presented in an intelligible fashion and written in standard English?

Reviewer #1: Yes

Reviewer #2: Yes

Reviewer #3: Yes

Reviewer #1: Thank you so much for providing me the opportunity to review the paper titled, “Sexual and Reproductive Health Challenges Among Street Adolescents in Sylhet City, Bangladesh: A Cross-Sectional Study”. The paper has analyzed an interesting issue, focusing on sexual reproductive health of street adolescents. The paper is well-written, including background, methodology, and results. However, I am suggesting some minor modification to the paper to make it more readable.

First, why the authors have chosen Sylhet city as study site, need explanation. As the case of Sylhet city might not be identical with other cities in Bangladesh regarding adolescent’s experience. It would be better to describe the context of Sylhet city in relation to other cities in Bangladesh.

Second, in the discussion section, the authors described and compared the results with other studies. However, they did not report convincingly the implications for policy and practice. For example, female adolescents are exposed to more sexual abuse than their male counterpart. So what? What policies and practices can be adopted in the family, community or state level to reduce risk and protect them? What is the implication for broader audiences other than the city studied or beyond Bangladesh? What are the socio-cultural barriers for not using SRH service?

Third, the authors used inferential statistics with a convenience sampling. They should report the reasons for not using probability sampling.

Fourth, there is an inconsistency in reporting results. (line no 146-47) Longer duration of street stay was also associated with 174 increased odds of abuse (OR: 1.14, 95% CI: 1.02-1.27, p=0.022) (Table 5). ) (Line 229-231) We found that adolescent girls with a younger age and a shorter duration of stay on the street were 230 significantly more likely to be sexually abused/ Which one is correct?

End.

Reviewer #2: I think the authors rightly addressed that the street adolescents who engage in early sexual activity, have multiple partners, and are at high risk of sexual abuse and exploitation and tried to explore the significance of this issue in Sylhet region.

Reviewer #3: This manuscript addresses a significant and underexplored topic in public health. The objective to understand the sexual and reproductive health (SRH) needs, practices, and challenges of the study population is highly relevant and commendable. However, major methodological and analytical concerns currently preclude the acceptance of the manuscript in its present form. Addressing these issues is essential.

Major Comments:

• Sample Size Justification: The sample size was calculated using the prevalence of 'sexual exposure' (71.6%). However, the study's primary objective is to understand a broad range of SRH needs, practices, and challenges, which are not considered for the sample size calculation. Please clarify the rationale for the sample size calculation.

• Risk of Investigator Bias: The description of data collection is unclear. The term "researcher" is not defined, and the involvement of the investigator's wife is noted. As the study team was fully aware of the research objectives, their direct role in conducting sensitive face-to-face interviews introduces a significant risk of observer and interviewer bias.

• Analytical Approach and Statistical Rigor: The analytical strategy requires review. The stratification of results in Tables 1, 3, and 4 by participant sex is not clearly justified by the study objectives. If sex comparison is not a central aim, this stratification is unnecessary.

• The measuring "sexual abuse" are not clearly described. The construction of the multivariate model (Table 5) needs further revised. The model uses current factors (e.g., age, marital status, education at the time of survey) to predict a lifetime history of sexual abuse. This creates a temporal fallacy where a current state is incorrectly presented as influencing a past event. The interpretation of current age as a protective factor (AOR: 0.62) against historical abuse is not logically correct. Similar to other factors as well.

**Do you want your identity to be public for this peer review?** For information about this choice, including consent withdrawal, please see our Privacy Policy

Reviewer #1: **Yes:**  Mohammad Shahjahan Chowdhury

Reviewer #2: No

Reviewer #3: **Yes:**  Faisal Ahmmed

---

## [Author Response · Author response to Decision Letter 1]

9 Nov 2025

Dear editor,

First of all, we would like to express our sincere gratitude for considering our work for review. We are thankful to the editors and reviewers for their valuable comments. We believe these comments have been instrumental in improving our manuscript quality. Herein, we presented a point-by-point response to the points raised by the reviewers. Any modification deemed necessary was made accordingly. An explanation was given when no modification was required or possible. Please note that we have replaced that S1 File with an updated version. Previously, we accidentally submitted an older version of the questionnaire in the S1 File. Each response to the reviewer’s comments are provided immediately below the comment.

Regards,

Dr. Md Golam Abbas, PhD

Associate Professor and Head,

Department of Occupational and Environmental Health,

National Institute of Preventive and Social Medicine,

Mohakhali, Dhaka 1212, Bangladesh.

Reviewer 1:

Thank you so much for providing me the opportunity to review the paper titled, “Sexual and Reproductive Health Challenges Among Street Adolescents in Sylhet City, Bangladesh: A Cross-Sectional Study”. The paper has analyzed an interesting issue, focusing on sexual reproductive health of street adolescents. The paper is well-written, including background, methodology, and results. However, I am suggesting some minor modification to the paper to make it more readable.

Authors’ response: Thank you very much for your time in reading our manuscript and providing your valuable comments.

First, why the authors have chosen Sylhet city as study site, need explanation. As the case of Sylhet city might not be identical with other cities in Bangladesh regarding adolescent’s experience. It would be better to describe the context of Sylhet city in relation to other cities in Bangladesh.

Authors’ response: Thank you. We added the necessary details and explanation in the introduction section. Please check lines 88 to 97.

Second, in the discussion section, the authors described and compared the results with other studies. However, they did not report convincingly the implications for policy and practice. For example, female adolescents are exposed to more sexual abuse than their male counterpart. So what? What policies and practices can be adopted in the family, community or state level to reduce risk and protect them? What is the implication for broader audiences other than the city studied or beyond Bangladesh? What are the socio-cultural barriers for not using SRH service?

Authors’ response: Thank you again for the nice suggestions. We added details on policy implications throughout the discussion where absent. Also, in lines 253 to 258 we added a separate paragraph discussing the importance of updating existing policy regarding children in Bangladesh.

Third, the authors used inferential statistics with a convenience sampling. They should report the reasons for not using probability sampling.

Authors’ response: Thank you for the comment. We have discussed this limitation in the limitations section of the manuscript. The sentences added were: “The study used convenience sampling as there was no updated information on the number of street children currently living in the city. Although this may limit the generalizability of the findings, to ensure variations in sampling, we collected data from different parts of the city where street children mostly stayed at night.” Please check lines 261 to 264.

Fourth, there is an inconsistency in reporting results. (line no 146-47) Longer duration of street stay was also associated with 174 increased odds of abuse (OR: 1.14, 95% CI: 1.02-1.27, p=0.022) (Table 5). ) (Line 229-231) We found that adolescent girls with a younger age and a shorter duration of stay on the street were 230 significantly more likely to be sexually abused/ Which one is correct?

Authors’ response: Thank you for the comment. Actually, “174” and “230” are line numbers displayed in the text; these are not part of the paragraph. Probably, it got mixed with during the formatting in the pdf built that was built within the submission system.

Reviewer 2:

I think the authors rightly addressed that the street adolescents who engage in early sexual activity, have multiple partners, and are at high risk of sexual abuse and exploitation and tried to explore the significance of this issue in Sylhet region.

Authors’ response: Thank you very much for your time in reading our manuscript and providing your valuable comments.

Reviewer 3:

This manuscript addresses a significant and underexplored topic in public health. The objective to understand the sexual and reproductive health (SRH) needs, practices, and challenges of the study population is highly relevant and commendable. However, major methodological and analytical concerns currently preclude the acceptance of the manuscript in its present form. Addressing these issues is essential.

Authors’ response: Thank you very much for your time in reading our manuscript and providing your valuable comments.

Major Comments:

• Sample Size Justification: The sample size was calculated using the prevalence of 'sexual exposure' (71.6%). However, the study's primary objective is to understand a broad range of SRH needs, practices, and challenges, which are not considered for the sample size calculation. Please clarify the rationale for the sample size calculation.

Authors’ response: Thank you. As there were a few quantifiable reports available on the different objectives, we chose to use the proportion of sexual experience by the street children in a similar study to calculate the sample size in our study. We added this statement in the sample size calculation subsection of the method section as well. Please check lines 109 to 118.

• Risk of Investigator Bias: The description of data collection is unclear. The term "researcher" is not defined, and the involvement of the investigator's wife is noted. As the study team was fully aware of the research objectives, their direct role in conducting sensitive face-to-face interviews introduces a significant risk of observer and interviewer bias.

Authors’ response: Thank you for the comment. We have replaced the term “researcher” with principal investigator, as he was the intended person to be mentioned here. We acknowledge the limitation of the investigator's bias in the limitations section. Please check lines 265 to 268.

• Analytical Approach and Statistical Rigor: The analytical strategy requires review. The stratification of results in Tables 1, 3, and 4 by participant sex is not clearly justified by the study objectives. If sex comparison is not a central aim, this stratification is unnecessary.

Authors’ response: Thank you for your concern. Actually, as this study was of a descriptive nature, the main intention was to describe the sexual and reproductive health experiences, needs, and challenges to some extent. Hence, the analyses presented to was shed some light on the interesting differences that can be interpreted. For example, we noted that based on the participant’s sex, our questionnaire had to include two separate parts for the reproductive questions. Also, characteristics and experiences of the respondents varied by sex. Hence, the tables 1, 3 and 4 present the findings by participants’ sex, along with the overall findings. However, thanks to your comments, we noted that Table 1 missed the overall column. We added the “overall” column in Table 1 as well. Without the stratification by sex, the differences in characteristics would not have emerged, and that adolescent street girls experience more sexual exploitation would not have been discussed with recommendations.

• The measuring "sexual abuse" are not clearly described. The construction of the multivariate model (Table 5) needs further revised. The model uses current factors (e.g., age, marital status, education at the time of survey) to predict a lifetime history of sexual abuse. This creates a temporal fallacy where a current state is incorrectly presented as influencing a past event. The interpretation of current age as a protective factor (AOR: 0.62) against historical abuse is not logically correct. Similar to other factors as well.

Authors’ response: Thank you for raising this important point. Actually, regression modeling can be done and interpreted in various ways. As this study was not a cohort study, we certainly cannot “predict” the sexual abuse among the participants. Hence, we used the logistic regression analysis to find possible factors associated with sexual abuse. Due to the cross-sectional nature of the study, the factors only indicate the characteristics that might have influenced abusive sexual experience. It only highlights that researchers in future studies should consider these factors to find if there are any predictive or causal relationships. In the limitations section of the manuscript, we have mentioned this limitation of the study. Nevertheless, the stakeholders can look at the associated factors to enhance policy measures based on the identified factors as well.

Please note that we have described how “sexual abuse” was defined and presented to the participants during the interview in the methods section to remove ambiguity. Please check lines 131 to 137.

Editor Comments:

You used a semi-structured questionnaire for your research. This implies use of both qualitative and quantitative methods in your study. The qualitative methods and results for your study need additional elaboration. The Equator Network guideline for reporting the results of qualitative health research provides a useful guideline for describing the results of studies like yours.

O'Brien BC, Harris IB, Beckman TJ, Reed DA, Cook DA. Standards for reporting qualitative research: a synthesis of recommendations. Acad Med. 2014;89(9):1245-1251.

https://www.equator-network.org/reporting-guidelines/srqr/

Providing this additional information may be useful in addressing the concerns expressed by Reviewer 3 and, to a lesser extent, Reviewer 1.

While elaborating on the qualitative aspects of your study, I would like you to elaborate on the following items:

• methods for collecting and interpreting the qualitative data in your study,

• elaboration of the major themes developed, for example "sexual abuse",

• Justification for the inferential statistics used to analyze your qualitative data.

Authors’ response: Thank you for the comment. We are sorry for the confusion that arose because of using the term “semi-structured” before the questionnaire. We used the term because there were some open questions where the respondents had to answer freely, rather than choosing from options. For example, “age”, “number of siblings”, “earning source”, “specific sexual problems”. Our study did not include any qualitative components where participants were asked for any opinion or experience on any issue. Also, if we used a qualitative component, the data collection instruments would be “interview guidelines” or “unstructured questionnaire”.

---

## [Decision Letter · Decision Letter 1]

29 Dec 2025

Sexual and Reproductive Health Challenges Among Street Adolescents in Sylhet City, Bangladesh: A Cross-Sectional Study

PONE-D-25-30858R1

Dear Dr. Abbas,

We’re pleased to inform you that your manuscript has been judged scientifically suitable for publication and will be formally accepted for publication once it meets all outstanding technical requirements.

Kind regards,

Christina M. Roberts, M.D., M.P.H.

Academic Editor

PLOS One

Additional Editor Comments (optional):

Thank you for your responses to the comments from myself and the reviewers.

Reviewer 3 corrently identifies that this is a cross sectional study and you are unable to make causal inferences based on this data. Reviewer 3 also notes that you cannot address this limitation except by commenting on this limitation in your discussion. I think you are careful not to make causal statements in your manuscript and I think the first statement in yout limitation section correctly aknowledges this limitation.

"Limitations and Strengths

This study has several limitations. The cross-sectional design prevents the establishment of causal relationships."

Reviewers' comments:

Reviewer's Responses to Questions

**Comments to the Author**

Reviewer #2: All comments have been addressed

Reviewer #3: All comments have been addressed

2. Is the manuscript technically sound, and do the data support the conclusions?

Reviewer #2: Yes

Reviewer #3: Partly

3. Has the statistical analysis been performed appropriately and rigorously?

Reviewer #2: Yes

Reviewer #3: No

4. Have the authors made all data underlying the findings in their manuscript fully available?

Reviewer #2: Yes

Reviewer #3: Yes

5. Is the manuscript presented in an intelligible fashion and written in standard English?

Reviewer #2: Yes

Reviewer #3: Yes

Reviewer #2: (No Response)

Reviewer #3: The authors attempted to respond to the reviewer’s comments; however, it is not possible to fully address all of them at this stage, except through acknowledgment as study limitations. Thank you for the effort.

Specially, for any statistical regression analysis, two types of variables are essential: the dependent variable (e.g., sexual abuse) and independent variables (e.g., age, marital status, education). In epidemiology, exposure variables (independent variables) should occur before the outcome (dependent variable) to support causal interpretation. Although the data were collected at the time of the interview, any observed statistical association between variables, for example, marital status and sexual abuse can only be considered causal if the exposure (marital status) was measured before the occurrence of the outcome (sexual abuse). This temporal ordering should be maintained for all risk and protective factor analyses. However, it is unclear whether all exposure variables were collected prior to the occurrence of sexual abuse. I did not find clear text or procedure for the measurement of sexual abuse and exposure variables.

**Do you want your identity to be public for this peer review?** For information about this choice, including consent withdrawal, please see our Privacy Policy

Reviewer #2: **Yes:**  Prof Dr Md Siddiqur Rahman

Reviewer #3: No

---

## [Editor Report · Acceptance letter]

PONE-D-25-30858R1

PLOS One

Dear Dr. Abbas,

I'm pleased to inform you that your manuscript has been deemed suitable for publication in PLOS One. Congratulations! Your manuscript is now being handed over to our production team.

Kind regards,

on behalf of

Dr. Christina M. Roberts

Academic Editor

PLOS One